# The r’-Wave Algorithm: A New Diagnostic Tool to Predict the Diagnosis of Brugada Syndrome after a Sodium Channel Blocker Provocation Test

**DOI:** 10.3390/s23063159

**Published:** 2023-03-16

**Authors:** Giampaolo Vetta, Antonio Parlavecchio, Lorenzo Pistelli, Paolo Desalvo, Armando Lo Savio, Michele Magnocavallo, Rodolfo Caminiti, Anna Tribuzio, Alessandro Vairo, Diego La Maestra, Francesco Vetta, Giuseppe Dattilo, Francesco Luzza, Gianluca Di Bella, Roberta Rossini, Domenico Giovanni Della Rocca, Pasquale Crea

**Affiliations:** 1Cardiology Unit, Department of Clinical and Experimental Medicine, University of Messina, 98122 Messina, Italy; 2Cardiology Unit, Department of Emergency and Critical Care, Hospital S. Croce e Carle, 12100 Cuneo, Italy; 3Cardiology Division, Arrhythmology Unit, S. Giovanni Calibita Hospital, Isola Tiberina, 00186 Rome, Italy; 4Division of Cardiology, Cardiovascular and Thoracic Department, Città della Salute e della Scienza University Hospital of Turin, 10126 Turin, Italy; 5Faculty of Medicine and Surgery, Saint Camillus International University of Health Sciences, 00131 Rome, Italy; 6Heart Rhythm Management Centre, Postgraduate Program in Cardiac Electrophysiology and Pacing, Universitair Ziekenhuis Brussel-Vrije Universiteit Brussel, European Reference Networks Guard-Heart, 1090 Brussels, Belgium

**Keywords:** Brugada syndrome, electrocardiogram, β-angle, α-angle, r’-wave, r’-wave algorithm

## Abstract

A diagnosis of Brugada syndrome (BrS) is based on the presence of a type 1 electrocardiogram (ECG) pattern, either spontaneously or after a Sodium Channel Blocker Provocation Test (SCBPT). Several ECG criteria have been evaluated as predictors of a positive SCBPT, such as the β-angle, the α-angle, the duration of the base of the triangle at 5 mm from the r’-wave (DBT- 5 mm), the duration of the base of the triangle at the isoelectric line (DBT- iso), and the triangle base/height ratio. The aim of our study was to test all previously proposed ECG criteria in a large cohort study and to evaluate an r’-wave algorithm for predicting a BrS diagnosis after an SCBPT. We enrolled all patients who consecutively underwent SCBPT using flecainide from January 2010 to December 2015 in the test cohort and from January 2016 to December 2021 in the validation cohort. We included the ECG criteria with the best diagnostic accuracy in relation to the test cohort in the development of the r’-wave algorithm (β-angle, α-angle, DBT- 5 mm, and DBT- iso.) Of the total of 395 patients enrolled, 72.4% were male and the average age was 44.7 ± 13.5 years. Following the SCBPTs, 24.1% of patients (n = 95) were positive and 75.9% (n = 300) were negative. ROC analysis of the validation cohort showed that the AUC of the r’-wave algorithm (AUC: 0.92; CI 0.85–0.99) was significantly better than the AUC of the β-angle (AUC: 0.82; 95% CI 0.71–0.92), the α-angle (AUC: 0.77; 95% CI 0.66–0.90), the DBT- 5 mm (AUC: 0.75; 95% CI 0.64–0.87), the DBT- iso (AUC: 0.79; 95% CI 0.67–0.91), and the triangle base/height (AUC: 0.61; 95% CI 0.48–0.75) (*p* < 0.001), making it the best predictor of a BrS diagnosis after an SCBPT. The r’-wave algorithm with a cut-off value of ≥2 showed a sensitivity of 90% and a specificity of 83%. In our study, the r’-wave algorithm was proved to have the best diagnostic accuracy, compared with single electrocardiographic criteria, in predicting the diagnosis of BrS after provocative testing with flecainide.

## 1. Introduction

Brugada syndrome (BrS), first introduced as a clinical entity in 1992, is a genetic disorder that increases the risk of sudden cardiac death secondary to ventricular tachycardia and ventricular fibrillation in patients without structural heart disease [1]. Brugada syndrome has an autosomal dominant transmission mode with variable penetrance involving several genes [1]. SCN5A, which encodes for the cardiac sodium channel α-subunit, is the most frequent mutation, occurring in 20–30% of patients [2]. Diagnosis of BrS is made by electrocardiogram (ECG) when a Brugada type 1 pattern is present either spontaneously or after a Sodium Channel Blocker Provocation Test (SCBPT) [2]. The indications for an SCBPT are an ECG that is suggestive, but not diagnostic, for BrS, and at least one of the following: symptoms (e.g., syncope, aborted cardiac arrest, sustained ventricular arrhythmias), a family history of BrS, or a family history of sudden cardiac death [2]. An SCBPT requires hospital admission and is not without risks [3]. Knowing the a priori probability of a positive SCBPT could be helpful in the decision whether to perform such a test, possibly leading to a reduction in unnecessary SCBPTs [3]. Several ECG criteria have been evaluated as predictors of a positive SCBPT. Chevallier et al. proposed the β-angle and the α-angle formed between the rise of the S-wave and the fall of the r’-wave as a predictive criterion—with high diagnostic accuracy—for distinguishing patients with a suspicious ECG from patients with a BrS diagnosis after an SCPBT [4]. Moreover, Serra et al. proposed the duration of the base of the triangle at 5 mm from the r’-wave (DBT- 5 mm), the duration of the base of the triangle at the isoelectric line (DBT- iso), and the triangle base/height ratio as predictors of a positive SBCPT with high predictive ability [5]. However, the diagnostic accuracy of these criteria has been shown to vary significantly among studies, and to date, studies comparing them are lacking [6,7]. The aim of our study was to test all previously proposed ECG criteria in a large cohort study and to evaluate an r’-wave algorithm for predicting a type 1 ECG pattern after an SCBPT.

## 2. Materials and Methods

### 2.1. Study Population

We enrolled all patients who consecutively underwent SCBPT at the Cardiology Unit of the Policlinico Gaetano Martino in Messina from January 2010 to December 2015 in the test cohort and from January 2016 to December 2021 in the validation cohort. Patients with symptoms (e.g., syncope, interrupted cardiac arrest), suggestive ECG features (e.g., a type 2 ECG), or a family history of BrS or sudden cardiac death underwent SCBPTs in the absence of a diagnostic ECG (type 1 ECG). The baseline ECG included V_1_ and V_2_ leads placed at the level of the fourth, third, and second intercostal spaces (IV°ic, III°ic, and II°ic). Flecainide was used as a sodium channel blocker and was infused intravenously in boluses to a maximum of 2 mg/kg body weight. The flecainide test was discontinued where there was a diagnostic type 1 ECG pattern, occurrence of ventricular arrhythmias, or QRS enlargement ≥40%. The study complied with the Declaration of Helsinki and the local ethics committee protocol. The informed consent of the subjects was obtained.

### 2.2. Electrocardiographic Analysis

ECG studies were recorded at a paper speed of 25 mm/s and at a standard gain of 1 mV/cm. ECG recordings were extracted and analyzed using the AMEDTEC ECGpro data management system. Two independent cardiologists (G.V. and A.P.) reviewed and interpreted all ECG tracings, and discordance was resolved by consensus. Defined parameters were P, PR, QRS, QT, QTc duration, and the P, QRS, and T axes.

The β-angle, α-angle, DBT- 5 mm, DBT- iso and triangle base/height ratio were analyzed if the r’-wave was measurable. The r’-wave was considered measurable in leads V1 and V2 with an amplitude of >0.1 mV (=1 mm) above baseline and a descending part of the r’-wave of >0.1 mV [6].

The β-angle was measured as the angle formed between the r’-wave upslope (yellow dotted line) and downslope (red dotted line) (Figure 1A). The α-angle was measured as the angle between a vertical line (green dotted line) and the downslope of the r’-wave (red dotted line) (Figure 1A). The duration of the base of the triangle was calculated as the length (d) between the intersection of the upslope and downslope of the r’-wave and a horizontal line 5-mm from the apex of the r’-wave (DBT- 5 mm) (Figure 1B) and a horizontal line at the level of the isoelectric line (DBT- iso) (Figure 1C). The triangle base/height ratio was calculated as the ratio of the duration at the base of the triangle (d) to the height (h) of the r’-wave from the isoelectric line (Figure 1C). Measurements were performed from both leads V1 and V2 at the IV°ic, III°ic, and II°ic, and the major value of each criterion was considered. Mean values from different beats measurements were calculated for each lead and each patient.

### 2.3. Statistical Analysis

Categorical variables were presented as frequencies (%) and compared using Fisher’s exact test. Continuous data were evaluated for normal distribution using histograms and the Kolmogorov–Smirnov test and expressed as mean ± standard deviation, or median (inter-quartile range) in the case of non-normal distribution. Comparisons were performed using an unpaired two-tailed t-test in case of normal distribution; otherwise, the Mann–Whitney U test was used. Receiver operating characteristic (ROC) curve analysis and the area under the ROC curve (AUC) were used to evaluate the diagnostic accuracy of ECG predictors of a positive SCBPT in the test and validation cohorts. AUCs were compared using the DeLong method. The McNemar test was used to assess lack of agreement with the gold standard (the flecainide provocative test) [8]. All ECG criteria that showed AUC >0.75 in the test cohort were included in the r’-wave algorithm development, and Youden’s index was used to evaluate the optimal cut-off value. The inter-observer and intra-observer agreement for the assessment of ECG predictors was assessed in 30 randomly selected patients in the validation cohort. A *p* value of less than 0.05 was considered significant. Statistical analysis was performed using SPSS Statistics (version 25.0, IBM Corporation, Armonk, NY, USA).

## 3. Results

### 3.1. Overall Population

We enrolled 395 patients, 72.4% of whom were male and whose average age was 44.7 ± 13.5 years.

The main indications for an SCBPT were a family history of BrS (49.5%), followed by a suspicious ECG (24.2%), a family history of sudden cardiac death (19.2%) and symptoms (syncope and a history of sustained ventricular tachycardia in the absence of structural heart disease) (7.1%). The baseline characteristics of the study population are shown in Table 1. SCBPTs were positive in 95 patients (24.1%) and negative in the remaining 75.9% (n = 300). Among patients with a positive SCBPT, the indication for an SCBPT was more often a suspicious ECG (32.6% vs. 21.7%; *p* = 0.03) and less frequently a family history of sudden cardiac death (6.3% vs. 23.3%; *p* < 0.0001). The dose (mg) of Flecainide administered was significantly lower in patients with a positive SCBPT (100 ± 12 vs. 130 ± 15; *p* < 0.0001).

### 3.2. Test Cohort 

We enrolled 198 patients in the test cohort, 70.2% (n = 139) of whom were male and whose average age was 44.5 ± 13.3 years. SCBPTs were positive in 24.2% (n = 48) of patients and negative in the remaining 75.8% (n = 150). The dose (mg) of Flecainide administered was lower in patients with a positive SCBPT (101 ± 11 vs. 129 ± 14; *p* < 0.0001). No other statistically significant differences between positive- and negative-SCBPT patients were elucidated concerning baseline characteristics and test indications (Table 2). In patients with a positive SCBPT, the r’-wave was most frequently measurable in at least one V1–V2 lead at the level of the IV° (18.8% vs. 4.0%; *p* = 0.001), III° (31.3% vs. 16.7%; *p* = 0.029), and II° intercostal spaces (66.7% vs. 22.7%; *p* < 0.0001) (Table 3). Patients with a positive SCBPT had a higher β-angle (53.1° vs. 34.3°; *p* < 0.0001), α-angle (41.4° vs. 25.3°; *p* < 0.0001), DBT- 5 mm (247.5 ms vs. 149 ms; *p* < 0.0001), DBT- iso (107.9 ms vs. 74.6 ms; *p* < 0.0001), and triangle base/height (1.4 vs. 1.1; *p* = 0.03) in comparison with negative-SCBPT patients (Table 3). ROC analysis showed that the AUC of the β-angle (AUC: 0.85; 95% CI 0.75–0.95), α-angle (AUC: 0.83; 95% CI 0.73–0.93), DBT- 5 mm (AUC: 0.83; 95% CI 0.72–0.94), and DBT- iso (AUC: 0.82; 95% CI 0.71–0.92) were significantly higher than that of the triangle base/height (AUC: 0.68; 95% CI 0.55–0.81) (*p* < 0.001), but no statistically significant difference between them was elucidated (*p* > 0.05) (Figure 2). Therefore, the β-angle, α-angle, DBT- 5 mm, and DBT- iso were all included in the development of the diagnostic algorithm. According to Youden’s index, the best diagnostic accuracy for the prediction of a BrS diagnosis after an SCBPT was obtained with a cut-off value of ≥40° for the β-angle (sensitivity 84.4%; specificity 80.0%) and ≥24° for the α-angle (sensitivity 93.8%; specificity 71.4%), and with a cut-off value of ≥120 ms for DBT- 5 mm (sensitivity 100.0%; specificity 73.5%) and ≥80 ms for DBT- iso (sensitivity 87.5%; specificity 74.3%). Therefore, we developed the r’-wave algorithm (Figure 3), which showed the best diagnostic accuracy on ROC analysis (AUC: 0.93; 95% CI 0.86–0.99) compared with the other criteria (*p* < 0.01) (Figure 4). According to Youden’s index, a cut-off value of ≥2 for the r’-wave algorithm showed the highest accuracy, with a sensitivity of 100.0% and specificity of 76.7%, for prediction of a BrS diagnosis after an SCBPT (Appendix A). 

### 3.3. Validation Cohort 

We enrolled 197 patients in the validation cohort, 74.6% (n = 147) of whom were male and whose average age was 44.9 ± 13.7 years. The Flecainide test was positive in 23.9% (n = 47) and negative in 76.1% (n = 150) of patients. The dose of Flecainide administered was lower in patients with a positive SCBPT compared with negative SCBPT (102 ± 12 vs. 128 ± 15; *p* < 0.0001). In patients with a positive SCBPT, the indication leading to the SCBPT was more often a suspicious ECG (38.3% vs. 20%, *p* = 0.01) and less frequently a family history of sudden cardiac death (0% vs. 36.7%, *p* < 0.0001). No other statistically significant differences between positive- and negative-SCBPT patients as regards baseline characteristics and test indications were elucidated (Table 4). In patients with a positive SCBPT, the r’-wave was most frequently measurable in at least one V1–V2 lead at the level of the IV° (27.7% vs. 6.0%; *p* < 0.0001), III° (59.6% vs. 18.0%; *p* < 0.0001), and II°ic (70.2% vs. 22.7%; *p* < 0.0001) (Table 5). Patients with a positive test had a higher β-angle (50.4° vs. 32.9°; *p* < 0.0001), α-angle (37.6° vs. 24.6°; *p* < 0.0001), DBT- 5 mm (189.2 ms vs. 115.4 ms; *p* < 0.0001), DBT- iso (112.0 ms vs. 69.8 ms; *p* < 0.0001), and triangle base/height (1.3 vs. 1.1; *p* = 0.03) in comparison with negative-SCBPT patients (Table 5). ROC analysis showed that the AUC of the r’-wave algorithm (AUC: 0.92; CI 0.85–0.99) was significantly better than the AUC of the β-angle (AUC: 0.82; 95% CI 0.71–0.92), α-angle (AUC: 0.77; 95% CI 0.66–0.90), DBT- 5 mm (AUC: 0.75; 95% CI 0.64–0.87), DBT- iso (AUC: 0.79; 95% CI 0.67–0.91), and triangle base/height (AUC: 0.61; 95% CI 0.48–0.75) (*p* < 0.001), making the r’-wave algorithm the best predictor of a BrS diagnosis after an SCBPT (Figure 5). The r’-wave algorithm with a cut-off value of ≥2 showed a sensitivity of 90% and a specificity of 83%, as shown in Appendix A.

### 3.4. Intra- and Inter-Observer Variability

Measurements of the β-angle, α-angle, DBT- 5 mm, DBT- iso, and triangle base/height showed an excellent intra-observer [ICC 0.984 (95% CI: 0.927–0.999; *p* < 0.001); ICC 0.985 (95% CI: 0.938–0.999; *p* < 0.001); ICC 0.857 (95% CI: 0.637–0.949; *p* < 0.001); ICC 0.950 (95% CI: 0.860–0.983; *p* < 0.001); and ICC 0.940 (95% CI: 0.890–0.980; *p* < 0.001), respectively] and inter-observer agreement [ICC 0.911 (95% CI: 0.687–0.988; *p* < 0.001); ICC 0.945 (95% CI: 0.746–0.977; *p* < 0.001); ICC 0.876 (95% CI: 0.654–0.931; *p* < 0.001); ICC 0.922 (95% CI: 0.825–0.999; *p* < 0.001); and ICC 0.865 (95% CI: 0.657–0.976; *p* < 0.001), respectively].

## 4. Discussion

BrS is an inherited heart disease caused by the inactivation of sodium channels in the right ventricle, which leads to the characteristic Brugada type 1 ECG pattern at the right precordial leads [9]. Patients with BrS may present with syncope, polymorphic ventricular tachycardia or ventricular fibrillation leading to sudden cardiac death [10]. A BrS diagnosis is based on the presence of a type 1 ECG pattern, either spontaneously or n a SCBPT performed in cases where there is suspicion related to symptoms, a type 2 ECG, or a family history of BrS or sudden cardiac death. An SCBPT requires hospitalization and is not risk-free, thus a priori determination of the probability of a positive SCBPT is necessary to avoid unnecessary provocative tests [3]. The ECG is the diagnostic test of reference in BrS, and a thorough analysis of r’ waves in V1–V2 leads can be crucial to differentiate benign ECG patterns from Brugada type 2 patterns [4,11,12]. Indeed, the r’-wave increases in size in patients with BrS, and this can be explained by two hypotheses. The first asserts that the slowed r’-wave results from an abnormal transmural gradient of right ventricular outflow tract (RVOT) repolarization, with a heterogeneous reduction in the duration and dome of epicardial action potentials and unaltered endocardial action potentials causing ST-segment elevation [13]. The second hypothesis states that ST-segment elevation results from delayed RVOT activation, resulting in prolonged depolarization time [13]. Placing the V_1_ and V_2_ leads in more cranial positions (III° and II°ic) increases sensitivity due to the variable anatomical correlation between the RVOT and the right precordial leads at IV°ic, as also shown in our study [14]. Several ECG criteria have been evaluated as predictors of a positive SCBPT, including the β-angle, the α-angle, the DBT- 5 mm, the DBT- iso, and the triangle base/height ratio, but these criteria showed different sensitivity and specificity values between different studies with different cut-offs, as shown in Appendix A [4,5,6,7,15]. Among these criteria, the β-angle is the most widespread and has been evaluated in several studies, but always with a high variability of sensitivity and specificity and without identifying an optimal cut-off [4,6,15]. This variability is due to the fact that the criteria were evaluated in small studies, and not all studies included the SCBPT as the gold standard. In our group’s recent meta-analysis, we showed that the β-angle maintained good diagnostic accuracy between studies (AUC: 0.75; 95% CI: 0.71–0.78), but we also confirmed high heterogeneity between them [16]. To the best of our knowledge, this is the first study comparing all previously studied criteria on a large population sample with the SCBPT as the gold standard. The β-angle, α-angle, DBT- 5 mm, and DBT- iso showed higher diagnostic accuracy than the triangle base/height. The r’-wave algorithm (developed from the β-angle, α-angle, DBT- 5 mm, and DBT- iso) proved to be statistically significantly superior for diagnostic accuracy compared with all other criteria in the test and validation cohorts. The use of a multi-criteria algorithm analyzing the r’-wave allows for higher diagnostic accuracy without suffering from the variability of single criteria highlighted in previous studies [4,5,6,7,15]. Furthermore, our study showed that the measurements of the β-angle, α-angle, DBT- 5 mm, DBT- iso and triangle base/height ratio showed an excellent intra-observer and inter-observer agreement. This new algorithm could be the first step in the process of developing new software that can automatically analyze the baseline ECG and select patients to undergo SCBT based on the pre-test probability predicted by the r’-wave algorithm. This study has some limitations. It is a single-centre study, which nevertheless has a large sample of patients. Furthermore, our population differs from that of other studies due to the less strict inclusion criteria. In fact, we enrolled all patients who underwent SCBPT at our centre because of suspicion of BrS and not only because of the presence of the Brugada type 2 ECG pattern, by analyzing the ECG criteria not only at the level of the IV° ic but also at the level of the III° and II° ic.

## 5. Conclusions

ECG signal analysis is a cornerstone for future development of computational ECG analysis. The identification of new algorithms, easily available and reproducible, may help in the development of software that may automatically analyze the baseline ECG, minimizing intra- and inter-observer variability, and favouring a standardization of the process. In our study, the r’-wave algorithm was proved to have the best diagnostic accuracy, compared with single electrocardiographic criteria, in predicting the diagnosis of BrS after provocative testing with Flecainide. Therefore, the r’-wave algorithm could assist in the choice of patients who should undergo SCBPT for a BrS diagnosis.

## Figures and Tables

**Figure 1 sensors-23-03159-f001:**
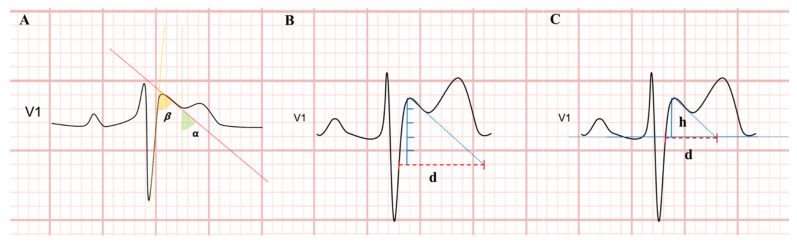
Measurement of β-angle and α-angle (**A**); duration of the base of the triangle at 5 mm from the r’-wave (**B**); duration of the base of the triangle at the isoelectric line and triangle base/height ratio (**C**).

**Figure 2 sensors-23-03159-f002:**
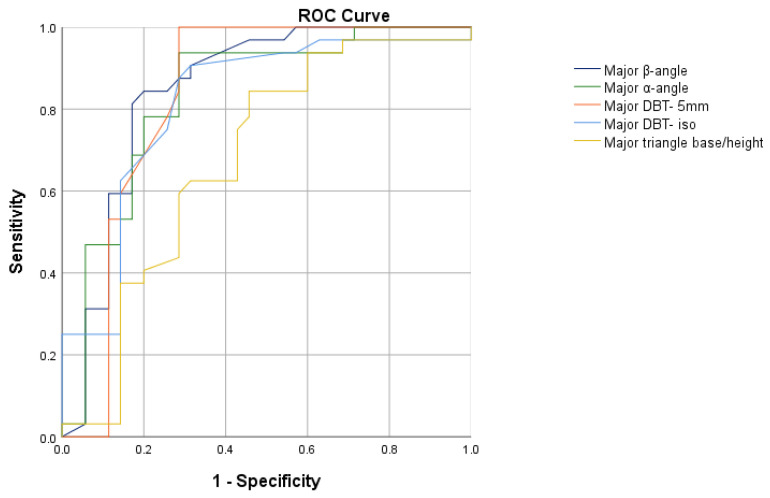
Receiver operating characteristic analyses of β-angle, α-angle, DBT- 5 mm, DBT- iso, and triangle base/height ratio to identify a BrS diagnosis after Sodium Channel Blocker Provocation Test in the test cohort.

**Figure 3 sensors-23-03159-f003:**
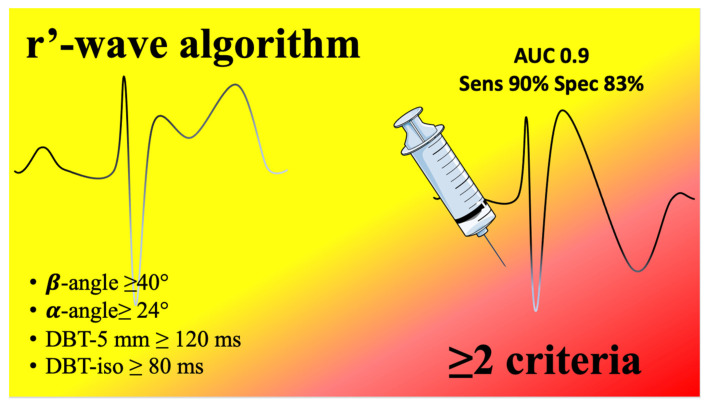
r’-wave algorithm illustration.

**Figure 4 sensors-23-03159-f004:**
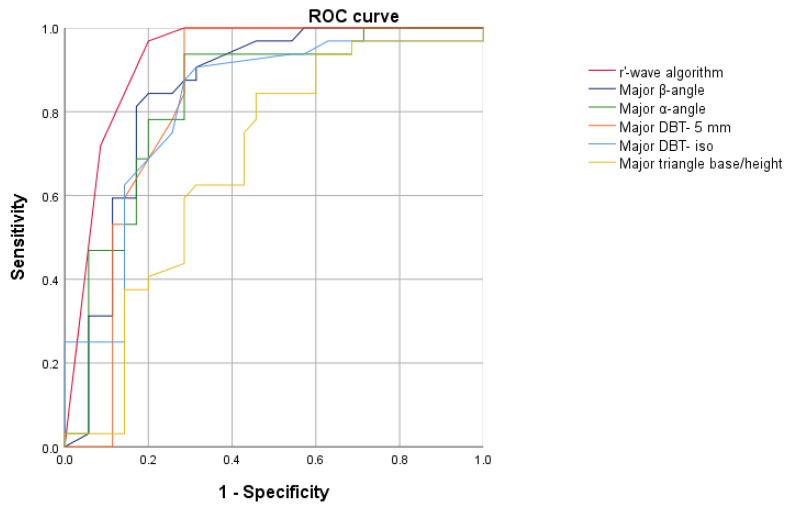
Receiver operating characteristic analyses of r’-wave algorithm, β-angle, α-angle, DBT- 5 mm, DBT- iso, and triangle base/height ratio to identify a BrS diagnosis after Sodium Channel Blocker Provocation Test in the test cohort.

**Figure 5 sensors-23-03159-f005:**
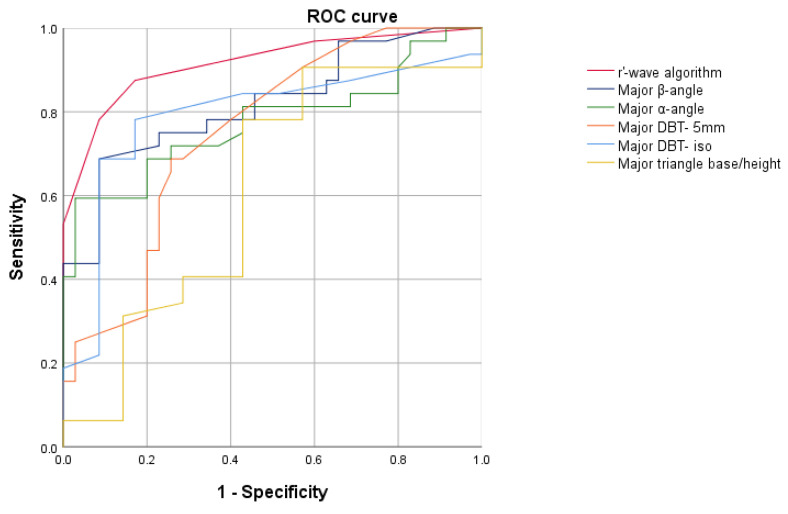
Receiver operating characteristic analyses of r’-wave algorithm, β-angle, α-angle, DBT- 5 mm, DBT- iso, and triangle base/height ratio to identify a BrS diagnosis after Sodium Channel Blocker Provocation Test in the validation cohort.

**Table 1 sensors-23-03159-t001:** Comparison of baseline clinical characteristics of the study population based on Brugada syndrome diagnosis after Sodium Channel Blocker Provocation Test.

	Overall (n = 395)	Negative SCBPT (n = 300)	Positive SCBPT (n = 95)	*p* Value
**Age, years**	44.7 ± 13.5	44.1 ± 13.6	46.8 ± 12.9	0.090
**Sex male, % (n)**	72.4 (286)	73.3 (220)	69.5 (29)	0.463
**LVEF, %**	58.7 ± 4.7	58.3 ± 5.3	59.3 ± 4.3	0.436
**Flecainide administered, mg**	122 ± 13	130 ± 15	100 ± 12	<0.0001
** *Indication for test* **				
**Suspicious ECG, % (n)**	24.2 (96)	21.7 (65)	32.6 (31)	0.03
**Symptoms, % (n)**	7.1 (28)	6.7 (20)	8.4 (8)	0.561
**Family history of BrS, % (n)**	49.5 (196)	48.3 (145)	52.6 (50)	0.465
**Family history of sudden cardiac death, % (n)**	19.2 (76)	23.3 (70)	6.3 (6)	<0.0001

**Table 2 sensors-23-03159-t002:** Comparison of baseline clinical characteristics of the test cohort based on Brugada syndrome diagnosis after Sodium Channel Blocker Provocation Test.

	Overall (n = 198)	Negative SCBPT (n = 150)	Positive SCBPT (n = 48)	*p* Value
**Age, years**	44.5 ± 13.3	44.1 ± 13.6	45.8 ± 12.2	0.455
**Sex Male, % (n)**	70.2 (139)	73.3 (110)	60.4 (29)	0.089
**LVEF, %**	58.9 ± 4.3	58.2 ± 5.2	59.6 ± 4.1	0.463
**Flecainide administered, mg**	123 ± 12	129 ± 14	101 ± 11	<0.0001
** *Indication for test* **				
**Suspicious ECG, % (n)**	24.2 (48)	23.3 (35)	27.1 (13)	0.598
**Symptoms, % (n)**	6.1 (12)	6.7 (10)	4.2 (2)	0.528
**Family history of BrS, % (n)**	59.1 (117)	60 (90)	56.3 (27)	0.646
**Family history of sudden cardiac death, % (n)**	10.6 (21)	10 (15)	12.5 (6)	0.624

**Table 3 sensors-23-03159-t003:** Comparison of ECG characteristics and criteria of the test cohort based on Brugada syndrome diagnosis after Sodium Channel Blocker Provocation Test.

	Overall (n = 198)	Negative SCBPT (n = 150)	Positive SCBPT (n = 48)	*p* Value
**Heart rate, bpm**	73.2 ± 14.2	73.7 ± 15.1	71.6 ± 11.2	0.368
**p-wave, ms**	104 ± 12.4	103.4 ± 12.4	106.2 ± 12.4	0.178
**PR interval, ms**	159.1 ± 23.6	157.9 ± 23.5	162.8 ± 23.9	0.209
**QRS duration, ms**	97.4 ± 7.4	97.2 ± 6.2	98.2 ± 10.4	0.410
**QT interval, ms**	386.5 ± 25.2	385.6 ± 25.9	389.5 ± 23.2	0.358
**QTc interval, ms**	418.8 ± 31.2	417.6 ± 33.2	422 ± 23.9	0.328
**P axis, °**	56.3 ± 21.4	57.2 ± 21.7	53.4 ± 20.4	0.291
**QRS axis, °**	41.4 ± 32.6	43.3 ± 34.3	36 ± 26.8	0.184
**T axis, °**	46.6 ± 21.5	45.80 ± 28.9	49.06 ± 23.2	0.360
** *Measurability of r’-wave at:* **				
**IV° ic, % (n)**	7.6 (15)	4 (6)	18.8 (9)	0.001
**III° ic, % (n)**	20.2 (40)	16.7 (25)	31.3 (15)	0.029
**II° ic, % (n)**	33.3 (66)	22.7 (34)	66.7 (32)	<0.0001
**Major** **β-** **angle**	43.3 ± 16.7	34.3 ± 14.9	53.1 ± 12.6	<0.0001
**Major** **α-** **angle**	32.9 ± 14.9	25.3 ± 12.5	41.4 ± 12.7	<0.0001
**Major DBT- 5 mm**	196.1 ± 126.2	149 ± 144.1	247.5 ± 76.9	<0.0001
**Major DBT- iso**	90.5 ± 34.5	74.6 ± 24.4	107.9 ± 35.8	<0.0001
**Major triangle base/height**	1.3 ± 0.7	1.1 ± 0.6	1.4 ± 0.7	0.03

**Table 4 sensors-23-03159-t004:** Comparison of baseline clinical characteristics of the validation cohort based on Brugada syndrome diagnosis after Sodium Channel Blocker Provocation Test.

	Overall (n = 197)	Negative SCBPT (n = 150)	Positive SCBPT (n = 47)	*p* Value
**Age, years**	44.9 ± 13.7	44.1 ± 13.6	47.9 ± 13.7	0.101
**Sex Male, % (n)**	74.6 (147)	73.3 (110)	78.7 (37)	0.459
**LVEF, %**	58.5 ± 4.6	58.2 ± 5.2	59.6 ± 4.1	0.463
**Flecainide administered, mg**	122 ± 11	128 ± 15	102 ± 12	<0.0001
** *Indication for test* **				
**Suspicious ECG, % (n)**	24.4 (48)	20 (30)	38.3 (18)	0.01
**Symptoms, % (n)**	8.1 (16)	6.7 (10)	12.8 (6)	0.182
**Family history of BrS, % (n)**	39.6 (78)	36.7 (55)	48.9 (23)	0.133
**Family history of sudden cardiac death, % (n)**	27.9 (55)	36.7 (55)	0 (0)	<0.0001

**Table 5 sensors-23-03159-t005:** Comparison of ECG characteristics and criteria of the validation cohort based on Brugada syndrome diagnosis after Sodium Channel Blocker Provocation Test.

	Overall (n = 197)	Negative SCBPT (n = 150)	Positive SCBPT (n = 47)	*p* Value
**Heart rate, bpm**	73.9 ± 15.1	73.7 ± 15.1	70.1 ± 14.8	0.150
**p-wave, ms**	103.8 ± 12.7	103.4 ± 12.4	105.2 ± 13.8	0.392
**PR interval, ms**	158.6 ± 24.7	157.9 ± 23.5	161 ± 28.3	0.451
**QRS duration, ms**	97.8 ± 9.1	97.2 ± 6.2	99.7 ± 15.1	0.104
**QT interval, ms**	385.8 ± 28.3	385.6 ± 25.9	386.3 ± 35.1	0.890
**QTc interval, ms**	416.2 ± 30.8	417.6 ± 33.2	411.9 ± 21.1	0.280
**P axis, °**	55.5 ± 20.1	57.2 ± 21.7	50.1 ± 12.1	0.06
**QRS axis, °**	40.1 ± 34.5	43.3 ± 34.3	30.6 ± 33.7	0.06
**T axis, °**	44.3 ± 20.8	45.8 ± 28.9	39.7 ± 20.1	0.079
** *Measurability of r’-wave at:* **				
**IV° ic, % (n)**	11.2 (22)	6 (9)	27.7 (13)	<0.0001
**III° ic, % (n)**	27.9 (55)	18 (27)	59.6 (28)	<0.0001
**II° ic, % (n)**	34 (67)	22.7 (34)	70.2 (33)	<0.0001
**Major** **β-** **angle**	41.4 ± 15.5	32.9 ± 8.7	50.4 ± 16.2	<0.0001
**Major** **α-** **angle**	30.8 ± 12.4	24.6 ± 6.9	37.6 ± 13.5	<0.0001
**Major DBT- 5 mm**	151.3 ± 85.9	115.4 ± 54.5	189.2 ± 97.1	<0.0001
**Major DBT- iso**	89.9 ± 46.9	69.8 ± 20.5	112 ± 57.1	<0.0001
**Major triangle base/height**	1.2 ± 0.7	1.1 ± 0.6	1.3 ± 0.7	0.03

## Data Availability

The data that support the findings of this study are available from the corresponding author upon reasonable request.

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
