# Peer review of "The r’-Wave Algorithm: A New Diagnostic Tool to Predict the Diagnosis of Brugada Syndrome after a Sodium Channel Blocker Provocation Test"

_sensors, 2023, doi:10.3390/s23063159_

Round 1
Reviewer 1 Report
All is as in the report.

Author Response
Response to Reviewer Comments
The title:
The r’-Wave Algorithm: a New Diagnostic Tool to Predict the Diagnosis of Brugada
Syndrome at Sodium Channel Blocker Provocation Test
By several authors
Submitted to Sensors MDPI type journal
The paper contains
Abstract, Keywords (4 words),
Introduction, Materials and Methods (Study Population, Electrocardiographic Analysis, Statistical analysis),
Results (Overall Population, Test Cohort, Validation Cohort, Intra and Inter-observer variability) Discussions,
Conclusion,
References (16 items) and it is written on 16 pages
I have read whole paper carefully and I have the following:
- A) General opinion
ECG signal analysis is a cornerstone for future development of computational ECG analysis. The identification of new algorithms, easily available and reproducible, may help in the development of software that may automatically analyses basal ECG minimizing intra and inter-observer variability and favoring a standardization of the process.
In this study, the r'-wave algorithm proved the best diagnostic accuracy over single electrocardiographic criteria in predicting the diagnosis of BrS on provocative testing with flecainide. Therefore, the r'-wave algorithm could guide the choice of patients who should undergo SCBPT for the BrS diagnosis.
R: We thank the reviewer for this valuable feedback.
- Remarks: Check the names of all the journals from the list of references including the abbreviated of ones; Also, check all tables and figures in the paper, etc,…Add maybe still some known book from this area.
R: We thank the reviewer for this valuable comment. We have amended the manuscript according to his advice.
Reviewer 2 Report
Several limitations were presented by the authors; however, the present study can be a starting point.
The paper should be carefully revised:
(Line: 37)
The r-wave algorithm is central to the presented paper, why not consider including it in the keywords?
(Line: 59)
End the Introduction with one paragraph describing the structure of the paper, section by section; use explicit notations, such as 'In Section 2, ...'
(Line: 216)
Supplemental Table 1 has a line number ‘included’ in column ‘Cut-off’
(Line: 244)
Table 4 value for ‘Overall’ does not correspond to the value presented in the text (line: 219)
(Line: 249)
Check for table 5 position.
(Line: 232-235)
Check values regarding the ones presented in table 5.
(Line: 332)
Check the name of the table, as well as its non-reference in the text.
Author Response
Response to Reviewer Comments
Several limitations were presented by the authors; however, the present study can be a starting point.
The paper should be carefully revised:
- (Line: 37) The r-wave algorithm is central to the presented paper, why not consider including it in the keywords?
R: We thank the reviewer for this valuable suggestion. As suggested we corrected Keywords:
“Keywords: Brugada syndrome; Electrocardiogram; β-angle; α-angle; r'-wave; r'-wave algorithm.”
- (Line: 59)
End the Introduction with one paragraph describing the structure of the paper, section by section; use explicit notations, such as 'In Section 2, ...'
R: We thank the reviewer for this valuable suggestion. But together with my co-authors, we considered that adding a paragraph introducing the structure of the article was not ideal.
- (Line: 216) Supplemental Table 1 has a line number ‘included’ in column ‘Cut-off’.
R: We thank the reviewer for this valuable suggestion. As suggested, we corrected:
“Supplemental Table 1. Cut-off values of the r'-wave algorithm in the Test cohort
|
Cut-off |
Sensitivity |
Specificity |
Youden Index |
|
≥1 |
100,0% |
48,6% |
0,486 |
|
≥2 |
100,0% |
76,7% |
0,767 |
|
≥3 |
83,2% |
80,0% |
0,632 |
|
≥4 |
71,9% |
91,4% |
0,633 |
”
- (Line: 244) Table 4 value for ‘Overall’ does not correspond to the value presented in the text (line: 219)
R: We thank the reviewer for this valuable suggestion. As suggested we fixed the mistake:
“Validation Cohort
We enrolled 197 patients in validation cohort, 74.6% (n=147) male with an average age of 44.9 ± 13.7 years. Flecainide test was positive in 23.9% (n=47) and negative in 76.1% (n=150) patients. ”
“
Table 4. Comparison of baseline clinical characteristics based on Brugada Syndrome diagnosis at Sodium Channel Blocker Provocation Test in the Validation cohort.
|
Overall (n=197) |
Positive-SCBPT (n=150) |
Negative-SCBPT (n=47) |
p value |
|
|
Age, years |
44. 9 ± 13.7 |
44.10 ± 13.60 |
47.85 ± 13.67 |
0.101 |
|
Sex Male, % (n) |
74.6 (147) |
73.3(110) |
78.7(37) |
0.459 |
|
LVEF, % |
58.5 ± 4.6 |
58.2 ± 5.2 |
59.6 ± 4.1 |
0.463 |
|
Flecainide administered, mg |
122 ± 11 |
128 ± 15 |
102 ± 12 |
0.0001 |
|
Indication for test |
||||
|
Suspicious ECG, % (n) |
24.4(48) |
20(30) |
38.3(18) |
0.01 |
|
Symptoms, % (n) |
8.1(16) |
6.7(10) |
12.8(6) |
0.182 |
|
Family history of BrS, % (n) |
39.6(78) |
36.7(55) |
48.9(23) |
0.133 |
|
Family history of sudden cardiac death, % (n) |
27.9(55) |
36.7(55) |
0(0) |
0.0001 |
“
- (Line: 249) Check for table 5 position.
R: We thank the reviewer for this valuable suggestion. We adjusted the position of table 5.
- (Line: 249) Check for table 5 position.
R: We thank the reviewer for this valuable suggestion. We adjusted the position of table 5.
- (Line: 232-235) Check values regarding the ones presented in table 5.
R: We thank the reviewer for this valuable suggestion. We fixed the incorrectly reported values:
“Patients with a positive test had a higher β-angle (50.4° vs 32.9°; p<0.0001), α-angle (37.6° vs 24.6°; p<0.0001), DBT- 5mm (189.2 ms vs 115.4 ms; p<0.0001), DBT- iso (112.0 ms vs 69.8 ms; p<0.0001) and triangle base/height (1.3 vs 1.1; p=0.03) in comparison to negative-SCBPT patients (Table 5).”
“
|
Overall (n=197) |
Negative-SCBPT (n=150) |
Positive-SCBPT (n=47) |
p value |
|
|
Heart rate, bpm |
73.9 ± 15.1 |
73.7 ± 15.1 |
70.1 ± 14.8 |
0.150 |
|
p-wave, ms |
103.8 ± 12.7 |
103.4 ± 12.4 |
105.2 ± 13.8 |
0.392 |
|
PR interval, ms |
158.6 ± 24.7 |
157.9 ± 23.5 |
161 ± 28.3 |
0.451 |
|
QRS duration, ms |
97.8 ± 9.1 |
97.2 ± 6.2 |
99.7 ± 15.1 |
0.104 |
|
QT interval, ms |
385.8 ± 28.3 |
385.6 ± 25.9 |
386.3 ± 35.1 |
0.890 |
|
QTc interval, ms |
416.2 ± 30.8 |
417.6 ± 33.2 |
411.9 ± 21.1 |
0.280 |
|
P axis, ° |
55.5 ± 20.1 |
57.2 ± 21.7 |
50.1 ± 12.1 |
0.06 |
|
QRS axis, ° |
40.1 ± 34.5 |
43.3 ± 34.3 |
30.6 ± 33.7 |
0.06 |
|
T axis, ° |
44.3 ± 20.8 |
45.8 ± 28. 9 |
39.7 ± 20.1 |
0.079 |
|
Measurability of r'-wave at: |
||||
|
IV° ic, % (n) |
11.2(22) |
6(9) |
27.7(13) |
<0.0001 |
|
III° ic, % (n) |
27.9(55) |
18(27) |
59.6(28) |
<0.0001 |
|
II° ic, % (n) |
34(67) |
22.7(34) |
70.2(33) |
<0.0001 |
|
Major β-angle |
41.4 ± 15.5 |
32.9 ± 8.7 |
50.4 ± 16.2 |
<0.0001 |
|
Major α-angle |
30.8 ± 12.4 |
24.6 ± 6.9 |
37.6 ± 13.5 |
<0.0001 |
|
Major DBT- 5mm |
151.3 ± 85.9 |
115.4 ± 54.5 |
189.2 ± 97.1 |
<0.0001 |
|
Major DBT- iso |
89.9 ± 46.9 |
69.8 ± 20.5 |
112 ± 57.1 |
<0.0001 |
|
Major triangle base/height |
1.2 ± 0.7 |
1.1 ± 0.6 |
1.3 ± 0.7 |
0.03 |
”
- (Line: 332) Check the name of the table, as well as its non-reference in the text.
R: We thank the reviewer for this valuable suggestion. We fixed the name of the table:
“Discussion
Several ECG criteria were evaluated as predictors of positive SCBPT such as β-angle, α-angle, DBT- 5 mm, DBT- iso and triangle base/height ratio but criteria showed different sensitivity and specificity values between different studies with different cut-offs, as shown in the Table 6 [3–7].”
|
Cut-off |
Article |
Sensitivity |
Specificity |
|
|
β-angle |
≥23° |
van der Ree et al.[5] |
77% |
62% |
|
≥23° |
Ohkubo et al.[6] |
100% |
54% |
|
|
≥23° |
Vetta et al.[16] |
83% |
65% |
|
|
≥23° |
Validation cohort |
97% |
23% |
|
|
≥36.8° |
van der Ree et al.[5] |
41% |
98% |
|
|
≥36.8° |
Serra et al.[3] |
86% |
95% |
|
|
≥36.8° |
Validation cohort |
85% |
54% |
|
|
≥38.6° |
van der Ree et al.[5] |
41% |
100% |
|
|
≥38.6° |
Serra et al.[3] |
85% |
96% |
|
|
≥38.6° |
Validation cohort |
76% |
77% |
|
|
≥58° |
Ohkubo et al.[6] |
23% |
100% |
|
|
≥58° |
van der Ree et al.[5] |
17% |
100% |
|
|
≥58° |
Vetta et al.[16] |
35% |
98% |
|
|
≥58° |
Chevallier et al.[4] |
79% |
83% |
|
|
≥58° |
Gottshalk et al.[7] |
60% |
78% |
|
|
≥58° |
Validation cohort |
42% |
100% |
|
|
DBT- 5mm |
≥160 ms |
Serra et al.[3] |
85% |
96% |
|
≥160 ms |
Validation cohort |
61% |
77% |
|
|
≥160 ms |
Gottshalk et al.[7] |
80% |
40% |
|
|
α-angle |
≥50° |
Chevallier et al.[4] |
71% |
79% |
|
≥50° |
Validation cohort |
22% |
100% |
|
|
DBT- iso |
≥60 ms |
Serra et al.[3] |
95% |
78% |
|
≥60 ms |
Validation cohort |
88% |
31% |
|
|
Triangle base/height ratio |
≥0.8 |
Serra et al.[3] |
82% |
92% |
|
≥0.8 |
Validation cohort |
84% |
43% |
“Table 6. Diagnostic characteristics of β-angle, α-angle, DBT- 5mm, DBT- iso and triangle base/height ratio cut-off values in studies“
Reviewer 3 Report
This manuscript investigates a study to test all previously proposed Electrocardiogram (ECG) criteria in a large cohort study and to evaluate a r’-wave algorithm to predict Type 1 pattern at Sodium Channel Blocker Provocation Test (SCBPT). Diagnosis of Brugada Syndrome (BrS) is made by the presence of Pattern Type 1 on the ECG, spontaneously or after a SCBPT. The a priori probability of a positive SCBPT, could be helpful in the decision whether to perform such a test, possibly leading to a reduction in unnecessary SCBPT. In this study, the r'-wave algorithm proved the best diagnostic accuracy over single electrocardio-graphic criteria in predicting the diagnosis of BrS on provocative testing with flecainide. The manuscript has some shortcomings. Some specific points/comments that need to be addressed are listed below.
1. Do not split the abstract into two paragraphs.
2. The introduction is too little and the background of the study is not fully presented.
3. The explanation of the terms "Sens" and "Spec" in line 42 should be consistent with the previous format.
4. The three dotted lines in Figure 1A are not clear enough to be obvious.
5. The description in lines 104-109 is repeated with the passage in 89-99.
6. Part 3 "Results" has too many paragraphs, some of which are too short. Also, tables are not presented in enough detail.
7. The poor arrangement of tables and pictures makes reading difficult.
In view of the above, the manuscript is not recommended for publication in its present form. The authors should modify the manuscript and address the above comments.
Author Response
Response to Reviewer Comments
This manuscript investigates a study to test all previously proposed Electrocardiogram (ECG) criteria in a large cohort study and to evaluate a r’-wave algorithm to predict Type 1 pattern at Sodium Channel Blocker Provocation Test (SCBPT). Diagnosis of Brugada Syndrome (BrS) is made by the presence of Pattern Type 1 on the ECG, spontaneously or after a SCBPT. The a priori probability of a positive SCBPT, could be helpful in the decision whether to perform such a test, possibly leading to a reduction in unnecessary SCBPT. In this study, the r'-wave algorithm proved the best diagnostic accuracy over single electrocardiographic criteria in predicting the diagnosis of BrS on provocative testing with flecainide. The manuscript has some shortcomings. Some specific points/comments that need to be addressed are listed below.
- Do not split the abstract into two paragraphs.
R: We thank the reviewer for this valuable suggestion. As suggested, we fixed the error:
“Abstract: Diagnosis of Brugada Syndrome (BrS) is made by the presence of Pattern Type 1 on the Electrocardiogram (ECG), spontaneously or after a Sodium Channel Blocker Provocation Test (SCBPT). Several ECG criteria were evaluated as predictors of positive SCBPT such as β-angle, α-angle, Duration of the Base of the Triangle at 5 mm from r′-wave (DBT- 5mm), Duration of the Base of the Triangle at the Isoelectric Line (DBT- iso) and triangle base/height ratio. The aim of our study was to test all previously proposed ECG criteria in a large cohort study and to evaluate a r’-wave algorithm for BrS diagnosis at SCBPT. We enrolled all patients consecutively underwent SCBPT with flecainide from January 2010 to December 2015 in the Test cohort and from January 2016 to December 2021 in the Validation cohort. We included the ECG criteria with the best di-agnostic accuracy at Test cohort in the development of the r'-wave algorithm (algorithm β-angle, α-angle, DBT- 5mm and DBT- iso.) We enrolled overall 395 patients, 72.4% male with an average age of 44.7 ± 13.5 years. At SCBPT test, 24.1% of patients (n=95) were positive and 75.9% (n=300) negative. At ROC analysis in the Validation cohort the AUC of r’-wave algorithm (AUC:0.92; CI 0.85 - 0.99) was significantly better than the AUC of β-angle (AUC: 0.82; 95% CI 0.71 - 0.92), α-angle (AUC: 0.77; 95% CI 0.66 - 0.90), DBT- 5mm (AUC: 0.75; 95% CI 0.64 - 0.87), DBT- iso (AUC: 0.79; 95% CI 0.67 - 0.91) and triangle base/height (AUC: 0.61; 95% CI 0.48 - 0.75) (p<0.001), resulting the best predictor of the BrS diagnosis at the SCBPT. The r'-wave algorithm with a cut-off value ≥ 2 showed a sensitivity of 90% and a specificity of 83%. In our study, the r'-wave algorithm proved the best diagnostic accuracy over single electrocardiographic criteria in predicting the diagnosis of BrS on provocative testing with flecainide.”
- The introduction is too little and the background of the study is not fully presented.
R: We thank the reviewer for this valuable suggestion. We expanded the introduction accordingly:
“1. Introduction
Brugada syndrome (BrS), first introduced as a clinical entity in 1992, is a genetic disorder that increases the risk of sudden cardiac death secondary to ventricular tachycardia and ventricular fibrillation in patients without structural heart disease [1]. Brugada syndrome has an autosomal dominant transmission mode with variable penetrance involving several genes [1]. SCN5A, which encodes for the cardiac sodium channel α-subunit, is the most frequent mutation occurring in 20-30% of patients [2]. Diagnosis of BrS is made by Electrocardiogram (ECG) when a Brugada Pattern Type 1 is present either spontaneously or after Sodium Channel Blocker Provocation Test (SCBPT) [2]. Indications to SCBPT are ECG suggestive, but not diagnostic, for BrS and at least one between: symptoms (e.g. syncope, aborted cardiac arrest, sustained ventricular arrhythmias), family history for BrS or family history sudden cardiac death [2]. SCBPT requires hospital admission and is not without risks [3]. The a priori probability of a positive SCBPT, could be helpful in the decision whether to perform such a test, possibly leading to a reduction in unnecessary SCBPT [3]. Several ECG criteria were evaluated as predictors of positive SCBPT. Chevallier et al. proposed the β-angle and α-angle included between the rise of the S-wave and the fall of the r′-wave as a predictive criterion to distinguish patients with suspicious ECG from patients with BrS diagnosis at SCPBT with high diagnostic accuracy [4]. Moreover, Serra et al. proposed the Duration of the Base of the Triangle at 5 mm from r′-wave (DBT- 5mm), Duration of the Base of the Triangle at the Isoelectric Line (DBT- iso) and triangle base/height ratio as predictors of positive SBCPT with high predictive ability [5]. However, the diagnostic accuracy of these criteria has been shown to vary significantly among studies, and to date, studies comparing them are absent [6,7]. Aim of our study was to test all previously proposed ECG criteria in a large cohort study and to evaluate a r’-wave algorithm to predict Type 1 pattern at SCBPT.”
- The explanation of the terms "Sens" and "Spec" in line 42 should be consistent with the previous format.
R: We thank the reviewer for this valuable suggestion. We modified accordingly:
“Abbreviations: AUC: Area Under the Curve; BrS: Brugada Syndrome; DBT- 5mm: Duration of the Base of the Triangle at 5 mm from r′-wave; DBT- iso: Duration of the Base of the Triangle at the Isoelectric Line; ECG: Electrocardiogram; Ic: intercostal space; LVEF: Left Ventricular Ejection Fraction; ROC: Receiver Operating Characteristic; RVOT: Right Ventricle Outflow Tract; SCBPT: Sodium Channel Blocker Provocation Test; Sens: Sensitivity; Spec: Specificity”
- The three dotted lines in Figure 1A are not clear enough to be obvious.
- The description in lines 104-109 is repeated with the passage in 89-99.
R: We thank the reviewer for this valuable suggestion. We modified it to make the figure explanation clearer and less repetitive:
Electrocardiographic Analysis
The β-angle is measured as the angle formed angle formed between the r’-wave upslope (yellow dotted line) and downslope (red dotted line)( Figure 1, A). The α-angle is measured as the angle between a vertical line (green dotted line) and the downslope of the r’-wave (red dotted line) (Figure 1, A). The duration at the base of the triangle was calculated as the length (d) between the intersection of the upslope and downslope of the r'-wave with a horizontal line 5-mm from the apex of the r'-wave (DBT- 5mm) (Figure 1, B) and with a horizontal line at the level of the isoelectric line (DBT- iso) (Figure 1, C). The ratio of base/height of the triangle was calculated as the ratio of the duration at the base of the triangle (d) and the height (h) of the r'-wave from the isoelectric line (Figure 1, C). Measurements were performed from both leads V1 and V2 at the IV°, III°, and II°ic and major value of each criterion was considered. Mean values from different beats measurements were calculated for each lead and each patient.
Figure 1. Measurement of β-angle and α-angle (A), Duration of the Base of the Triangle at 5 mm from r′-wave (B), Duration of the Base of the Triangle at the Isoelectric Line and triangle base/height ratio (C).”
- Part 3 "Results" has too many paragraphs, some of which are too short. Also, tables are not presented in enough detail.
R: We thank the reviewer for this valuable suggestion. We modified accordingly:“
- Results
Overall Population
We enrolled 395 patients, 72.4% male with an average age of 44.7 ± 13.5 years.
The main indication for SCBPT was family history of BrS (49.5%), followed by ECG suspect (24.2%), family history of sudden cardiac death (19.2%) and symptoms (syncope and history of sustained ventricular tachycardia in the absence of structural heart disease) (7.1%). Baseline characteristics of the population in study are shown in Table 1. SCBPT was positive in 95 patients (24.1%) and negative in the remaining 75.9% (n=300). Among patients with a positive SCBPT, the indication to SCBPT was more often a suspicious ECG (32.6% vs. 21.7%, p=0.03) and less frequently family history of sudden cardiac death (6.3% vs. 23.3%, p < 0.0001). Flecainide administered dose (mg) was significantly lower in patients with a positive SCBPT (100 ± 12 vs 130 ± 15; p<0.0001).
Test Cohort
We enrolled 198 patients in Test cohort, 70.2% (n=139) male, average age of 44.5 ± 13.3 years. SCBPT was positive in 24.2% (n=48) of patients, while resulted negative in the remaining 75.8% (n=150). Dose of administered flecainide (mg) was lower in patients with a positive SCBPT (101 ± 11 vs 129 ± 14; p<0.0001). No other statistically significant differences between positive- and negative- SCBPT patients were enlighten for what concerns baseline characteristics and test indications (Table 2). In patients with a positive-SCBPT, r’-wave was most frequently measurable in at least one V1-V2 lead at the level of the IV° (18.8% vs 4.0%; p=0.001), III° (31.3% vs 16.7%; p=0.029) and II° intercostal space (66.7% vs 22.7%; p<0.0001) (Table 3). Patients with a positive-SCBPT had a higher β-angle (53.1° vs 34.3°; p<0.0001), α-angle (41.4° vs 25.3°; p<0.0001), DBT- 5mm (247.5 ms vs 149 ms; p<0.0001), DBT- iso (107.9 ms vs 74.6 ms; p<0.0001) and triangle base/height (1.4 vs 1.1; p=0.03) in comparison to negative-SCBPT patients (Table 3). At ROC analysis, the AUC of β-angle (AUC:0.85; 95% CI 0.75 - 0.95), α-angle (AUC: 0.83; 95% CI 0.73 - 0.93), DBT- 5mm (AUC: 0.83; 95% CI 0.72 - 0.94), DBT- iso (AUC: 0.82; 95% CI 0.71 - 0.92) were significantly higher than triangle base/height (AUC: 0.68; 95% CI 0.55 - 0.81) (p<0.001) but no statistically significant difference between them was enlighten (p>0.05) (Figure 2). Therefore, β-angle, α-angle, DBT- 5mm and DBT- iso were all included in the development of a diagnostic algorithm. According to Youden index, the best diagnostic accuracy to predict BrS diagnosis at the SCBPT was allowed with the cut-off value ≥ 40° for β-angle (sensitivity 84.4%; specificity 80.0%) and ≥ 24° for α-angle (sensitivity 93.8%; specificity 71.4%), and with a cut-off value ≥ 120 ms for DBT- 5mm (sensitivity 100.0%; specificity 73.5%) and ≥ 80 ms for DBT- iso (sensitivity 87.5%; specificity 74.3%). Therefore, we developed the r’-wave algorithm (Figure 3), which showed the best diagnostic accuracy on ROC analysis (AUC: 0.93; 95% CI 0.86 - 0.99) compared to the other criteria (p<0.01) (Figure 4). According to the Youden index, the cut-off value of ≥ 2 for r’-wave algorithm showed the highest accuracy with a sensitivity of 100.0% and specificity 76.7% to predict the BrS diagnosis at the SCBPT (Supplemental Table 1).
Validation Cohort
We enrolled 197 patients in validation cohort, 74.6% (n=147) male with an average age of 44.9 ± 13.7 years. Flecainide test was positive in 23.9% (n=47) and negative in 76.1% (n=150) patients. The administered dose of flecainide was lower in patients with a positive SCBPT compared to negative SCBPT (102 ± 12 vs 128 ± 15; p<0.0001). In patients with a positive SCBPT, indication leading to SCBPT was more often a suspicious ECG (38.3% vs. 20%, p=0.01) and less frequently family history of sudden cardiac death (0% vs. 36.7%, p < 0.0001). No other statistically significant differences between positive- and negative- SCBPT patients in baseline characteristics and test indications were enlighten (Table 4). In patients with a positive SCBPT, r’-wave was most frequently measurable in at least one V1-V2 lead at the level of the IV° (27.7% vs 6.0%; p<0.0001), III° (59.6% vs 18.0%; p<0.0001) and II°ic (70.2% vs 22.7%; p<0.0001) (Table 5). Patients with a positive test had a higher β-angle (50.4° vs 32.9°; p<0.0001), α-angle (37.6° vs 24.6°; p<0.0001), DBT- 5mm (189.2 ms vs 115.4 ms; p<0.0001), DBT- iso (112.0 ms vs 69.8 ms; p<0.0001) and triangle base/height (1.3 vs 1.1; p=0.03) in comparison to negative-SCBPT patients (Table 5). At ROC analysis the AUC of r’-wave algorithm (AUC:0.92; CI 0.85 - 0.99) was significantly better than the AUC of β-angle (AUC: 0.82; 95% CI 0.71 - 0.92), α-angle (AUC: 0.77; 95% CI 0.66 - 0.90), DBT- 5mm (AUC: 0.75; 95% CI 0.64 - 0.87), DBT- iso (AUC: 0.79; 95% CI 0.67 - 0.91) and triangle base/height (AUC: 0.61; 95% CI 0.48 - 0.75) (p<0.001), resulting the best predictor of the BrS diagnosis at the SCBPT (Figure 5). The r'-wave algorithm with a cut-off value ≥ 2 showed a sensitivity of 90% and a specificity of 83%, as shown in Supplemental Table 2.
Intra and Inter-observer variability
Measurements of β-angle, α-angle, DBT- 5mm, DBT- iso and triangle base/height showed an excellent intra-observer [ICC 0.984 (95% CI: 0.927–0.999; p< 0.001); ICC 0.985 (95% CI: 0.938–0.999; p< 0.001); ICC 0.857 (95% CI: 0.637–0.949; p< 0.001); ICC 0.950 (95% CI: 0.860–0.983; p< 0.001); and 0.940 (95% CI: 0.890–0.980; p< 0.001), respectively] and inter-observer agreement [ICC 0.911 (95% CI: 0.687–0.988;p< 0.001); ICC 0.945 (95% CI: 0.746–0.977; p< 0.001); ICC 0.876 (95% CI: 0.654–0.931; p< 0.001); ICC 0.922 (95% CI: 0.825–0.999; p< 0.001); and 0.865 (95% CI: 0.657–0.976; p< 0.001), respectively].
“
- The poor arrangement of tables and pictures makes reading difficult.
R: We thank the reviewer for this valuable suggestion. We adjusted the arrangement of the figures and tables. In addition, we moved some tables to supplementary materials.
Round 2
Reviewer 3 Report
The authors have addressed my issues.